# A Bibliometric Analysis of *Cannabis* Publications: Six Decades of Research and a Gap on Studies with the Plant

**Cristiane B. D. Matielo** [1], **Deise S. Sarzi** [2], **Beatriz Justolin** [1], **Rafael P. M. Lemos** [1], **Flavio A. O. Camargo** [3] **and Valdir M. Stefenon** [1,*]

[1] Núcleo de Ecologia Molecular e Micropropagação de Plantas, Universidade Federal do Pampa—UNIPAMPA, São Gabriel 97307-020, Rio Grande do Sul, Brazil; crixdoliveira@gmail.com (C.B.D.M.); biasinjustolin@gmail.com (B.J.); rafael.matielo@unipampa.edu.br (R.P.M.L.)

[2] Instituto de Bioquímica Médica Leopoldo de Meis—CCS, Universidade Federal do Rio de Janeiro—UFRJ, Rio de Janeiro 21941-902, RJ, Brazil; deisesarzi@gmail.com

[3] Departamento de Solos, Faculdade de Agronomia, Universidade Federal do Rio Grande do Sul—UFRGS, Porto Alegre 91501-970, Rio Grande do Sul, Brazil; fcamargo@ufrgs.br

[*] Correspondence: valdirstefenon@unipampa.edu.br; Tel.: +55-55-3237-0851

**Abstract:** In this study we performed a bibliometric analysis focusing on the general patterns of scientific publications about *Cannabis*, revealing their trends and limitations. Publications related to *Cannabis*, released from 1960 to 2017, were retrieved from the Scopus database using six search terms. The search term "Genetics" returned 53.4% of publications, while "forensic genetics" and "traceability" represented 2.3% and 0.1% of the publications, respectively. However, 43.1% of the studies were not directly related to *Cannabis* and, in some cases, *Cannabis* was just used as an example in the text. A significant increase in publications was observed after 2001, with most of the publications coming from Europe, followed by North America. Although the term *Cannabis* was found in the title, abstract, or keywords of 1284 publications, we detected a historical gap in studies on *Cannabis*. We expect that increasing interest in this issue and the rise of new biotechnological advances will lead to the development of new studies. This study will help scientists identify overall research needs, detect the scientific areas in evidence concerning *Cannabis* studies, and find excellent centers of investigation for scientific interchange and collaboration.

**Keywords:** biotechnological advances; forensic science; hemp; genetics; marijuana; traceability

## 1. Introduction

*Cannabis sativa* L. is the most cultivated, trafficked, and consumed illicit drug in the world [1,2]. The United Nations classifies cannabis as all drugs derived from the plant *Cannabis sativa* (Cannabaceae) containing the substance Δ-9-tetrahydrocannabinol (THC). THC is the intoxicant compound of the plant and is responsible for the classification of *C. sativa* as an illicit drug. The plant also produces the cannabidiol (CBD), a compound that has been studied for pharmaceutical and medical purposes.

Aiming to increase the concentration of THC in the plant, several artificial hybrids such as "Royal AK47," "Sharksbreath," "Black Widow," "Haze Prata," "Kali Mist," and "Jack the Ripper" have been developed through genetic selection and breeding of *C. sativa* [3–5]. On the other hand, varieties such as "Cheungsam" were developed aiming at a THC/CBD ratio that is lower than 1.0 for the medicinal use of *Cannabis* [6], while strains such as "Félina 34," "Futura 77," "Kompolti," and "Carmagnola" were developed for cultivation aiming at the production of fibers [7]. Since the morphological differentiation

among all those varieties is very difficult, genetic strategies are necessary for their characterization with forensic purposes.

Forensic genetics arose as a result of the union between legal medicine and criminalistics and is most commonly linked to the use of human DNA in criminal investigations. However, the evolution of our society significantly enlarged its framework, and forensic genetics now covers a much wider range of purposes, providing subsidiary evidence in investigations involving cases such as biopiracy, bioterrorism, identification of fraudulent food composition, and identification of illicit drugs [8].

Besides forensic investigations aiming to differentiate drug and non-drug varieties of *Cannabis*, forensic genetics can be very useful for determining the geographic origin of seeds/plants, and for identification of traffic routes and illegal plantations. Molecular markers employed for genetic studies in *C. sativa* have shown that it is possible to correlate the diversity of gene pools with their geographical origin [9–12], suggesting that these biotechnological tools can be used for forensic purposes.

Taking into account the social impacts of *Cannabis* traffic and its abuse, as well as recent advances in biotechnological methods, it should be expected that scientific studies on forensic genetics related to this species would have significantly increased in recent decades. An increased number of systematic reviews and meta-analyses about the medical and psychological effects of *Cannabis* have been launched recently in the specialized literature [13–16]. However, review studies focusing on the general patterns of scientific researches of this species, revealing trends and limitations are lacking. Such analysis might help scientists and students to identify overall research needs, to detect the scientific areas in evidence concerning *Cannabis* studies, and to find reputable research centers for scientific interchange and collaboration.

Despite the importance of *Cannabis* for public health and security, no review or bibliometric study about the publications on this species is available in the specialized literature. In order to fill this gap, we intended to qualitatively scrutinize the patterns of scientific publications (research articles and review papers) related to *Cannabis* in the last six decades. In this study, we examined the main areas of interest of the researchers studying *Cannabis* around the world and the existing gaps concerning the current demands of research, with particular attention to forensic genetics.

## 2. Materials and Methods

### 2.1. Search Strategy

Scientific publications related to *Cannabis*, released from 1960 to 2017, were retrieved from the Scopus®(http://www.scopus.com) in June 2018. Scopus is a global multidisciplinary database with larger coverage compared to other online platforms such as Web of Science and the Scientific Electronic Library Online. Currently, it covers about 15,000 international peer-reviewed journals in the field of science and technology.

A previous search using the same parameters described below was also performed on Web of Science (https://clarivate.com/products/web-of-science/) and on the Scientific Electronic Library Online (SCIELO; www.scielo.org) platforms. When searching in different databases, specific contributions are usually found. However, the information returned from Web of Science and SCIELO in this study was also found in the SCOPUS results, without missing any publications, which confirms the larger coverage of the later database, as previously shown by Stefenon et al. [17].

As the information registered in SCOPUS was more complete and there were no data lost by excluding the Web of Science and SCIELO records, the study was performed using only data from the former platform. In addition, using data recorded from a single platform with such a wide coverage avoids introducing bias by including duplicated publications.

Only research articles and review papers were considered in the present study. Scientific notes and short communications reporting experimental studies were grouped into the research articles category.

The search arguments used were (i) *Cannabis* + Biochemical, (ii) *Cannabis* + Biology, (iii) *Cannabis* + Forensic genetics, (iv) *Cannabis* + Genetics, (v) *Cannabis* + Molecular markers and (vi) *Cannabis* +

Traceability. Using the generic name as search argument allowed us to assess publications about any of the species/sub-species, as well as their hybrids. The search was performed in the fields "Article title," "Abstract," and "Keywords." Since the search considers just the presence of the argument, and not its meaning, all recorded publications were individually checked in order to exclude studies unrelated to the topic.

*2.2. Overall Bibliometric Analysis*

The results were divided into six decades (1961–1970, 1971–1980, 1981–1990, 1991–2000, 2001–2010, and 2011–2017), by type of publication (research article or review paper), knowledge area (25 different areas according to the SCOPUS classification), and country of origin of the corresponding author.

The evolution in the number of publications was determined through the absolute number of publications and by computing the relative growth rate as RGR = $(\ln N_2 - \ln N_1)/(t_2 - t_1)$, where $N_2$ and $N_1$ are the cumulative number of publications in the years $t_2$ and $t_1$, respectively [18]. Graphics and statistical computations were performed using Microsoft Excel® spreadsheet software.

*2.3. Screening and Data Summarization*

Titles, abstracts, and objectives of the publications identified by the Scopus platform, as related to the search arguments, were independently screened and the data were summarized by two authors (C.B.D.M. and V.M.S.). The inclusion criteria were: (i) the organism related in the study, (ii) use for studies in *Cannabis*, and (iii) the principal focus for the study. Differences were resolved by discussion and consultation with a third author (R.P.M.L.).

**3. Results**

*3.1. Overall Results in the Scopus Platform*

Using the defined criteria, the search yielded a total of 1.284 publications, with prevalence of research articles over review papers (74.53% and 23.52% respectively). Scientific notes and short communications represented 0.85% and 1.09% of the total of publications, respectively, and given their nature were grouped with research articles (Table 1).

The search argument "*Cannabis* + Genetics" returned the largest number of publications (53.4%), followed by "*Cannabis* + Biochemical" (23.1%). The search arguments returning the lowest number of publications were "*Cannabis* + Forensic genetics" and "*Cannabis* + Traceability" (Table 1). The search arguments "*Cannabis* + Biochemical" and "*Cannabis* + Genetics" were the most important during all the investigated period, with a very similar number of publications returned for these two arguments from 1961 to 2000.

After 2001, the search argument "*Cannabis* + Genetics" appeared more often in the retrieved publications (Table 2). Publications with the search terms "*Cannabis* + Forensic genetics" were retrieved only after 2001, while the search terms "*Cannabis* + Traceability" only appeared in the last decade (Table 2).

The relative growth rate (Figure 1a) presented the same trend for the search arguments "*Cannabis* + Biochemical," "*Cannabis* + Biology," and "*Cannabis* + Genetics." The search argument "*Cannabis* + Molecular markers" increased after the year 1981, following the same tendency in the subsequent years. "*Cannabis* + Forensic genetics" revealed a significant growth in the two last decades, while "*Cannabis* + Traceability" basically lacked growth giving the quite low number of publications returned for this search argument (only two articles in the last decade).

**Table 1.** Number of scientific publications recorded from the Scopus platform for each search argument (*Cannabis* + . . . ) over the whole timespan in each knowledge area and in each continent.

| | Search Argument (*Cannabis* + . . . ) | | | | | |
| --- | --- | --- | --- | --- | --- | --- |
| | Biochemical | Biology | Forensic Genetics | Genetics | Molecular Markers | Traceability |
| Number of publications (%) | 297 (23.1%) | 210 (16.4%) | 29 (2.3%) | 686 (53.5%) | 60 (4.6%) | 2 (0.1%) |
| Number of research articles | 223 | 123 | 27 | 531 | 51 | 2 |
| Number of review papers | 71 | 79 | 1 | 142 | 9 | 0 |
| Number of notes | 0 | 2 | 1 | 8 | 0 | 0 |
| Number of short surveys | 3 | 6 | 0 | 5 | 0 | 0 |
| Knowledge areas (in %) [$] | | | | | | |
| Medicine | 46.80 | 45.97 | 81.48 | 67.74 | 46.67 | 0.50 |
| Neuroscience | 13.80 | 16.59 | 0.00 | 21.04 | 16.67 | 0.00 |
| Biochemistry, Genetics and Molecular Biology | 26.26 | 30.33 | 55.55 | 27.96 | 38.33 | 0.00 |
| Pharmacology, Toxicology and Pharmaceutics | 26.60 | 27.49 | 7.41 | 19.20 | 30.00 | 0.00 |
| Psychology | 4.04 | 5.21 | 0.00 | 11.37 | 3.33 | 0.00 |
| Agriculture and Biological Science | 13.47 | 14.69 | 7.41 | 8.75 | 23.33 | 0.00 |
| Social Science | 2.36 | 2.37 | 7.41 | 3.53 | 1.67 | 0.00 |
| Environmental science | 3.36 | 2.37 | 0.00 | 1.23 | 0.00 | 0.00 |
| Earth and Planetary Science | 0.00 | 0.00 | 0.00 | 0.00 | 0.00 | 0.50 |
| Immunology and Microbiology | 2.69 | 5.69 | 0.00 | 5.53 | 1.67 | 0.00 |
| Chemistry | 3.70 | 1.90 | 3.70 | 2.76 | 6.67 | 0.00 |
| Multidisciplinary | 2.36 | 3.79 | 0.00 | 0.31 | 0.00 | 0.00 |
| Arts and Humanities | 1.01 | 0.95 | 0.00 | 2.15 | 0.00 | 0.00 |
| Nursing | 0.00 | 0.95 | 3.70 | 1.23 | 0.00 | 0.00 |
| Chemical Engineering | 3.03 | 1.90 | 0.00 | 0.77 | 1.67 | 0.00 |
| Health Professional | 0.34 | 0.47 | 0.00 | 0.15 | 1.67 | 0.00 |
| Engineering | 1.01 | 0.47 | 0.00 | 0.00 | 0.00 | 0.00 |
| Material Science | 1.35 | 2.84 | 0.00 | 0.00 | 0.00 | 0.00 |
| Computer Science | 0.34 | 0.95 | 0.00 | 0.31 | 0.00 | 0.00 |
| Energy | 0.34 | 0.00 | 0.00 | 0.00 | 0.00 | 0.00 |
| Dentistry | 0.34 | 0.00 | 0.00 | 0.15 | 0.00 | 0.00 |
| Economics, Econometrics and Finance | 0.34 | 0.00 | 0.00 | 0.00 | 0.00 | 0.00 |
| Veterinary | 0.34 | 0.00 | 0.00 | 0.00 | 0.00 | 0.00 |
| Mathematics | 0.00 | 0.47 | 0.00 | 0.15 | 0.00 | 0.00 |

**Table 1.** *Cont.*

| | Search Argument (*Cannabis + . . .* ) | | | | | |
| | Biochemical | Biology | Forensic Genetics | Genetics | Molecular Markers | Traceability |
|---|---|---|---|---|---|---|
| Physics and Astronomy | 0.00 | 0.00 | 0.00 | 0.00 | 0.00 | 0.00 |
| Undefined | 0.00 | 0.00 | 0.00 | 0.15 | 0.00 | 0.00 |
| Publications by continent (in %) [$$] | | | | | | |
| Africa | 5.38 | 0.47 | 0.00 | 1.54 | 0.00 | 0.00 |
| Latin America | 3.03 | 2.84 | 3.70 | 2.15 | 3.33 | 0.00 |
| North America | 34.68 | 46.44 | 33.33 | 55.60 | 31.67 | 100.00 |
| Asia | 9.43 | 9.95 | 11.11 | 9.83 | 11.66 | 0.00 |
| Europe | 59.60 | 57.82 | 51.85 | 56.06 | 65.00 | 0.00 |
| Oceania | 2.02 | 4.27 | 7.41 | 8.49 | 5.00 | 0.00 |
| Undefined | 0.34 | 0.49 | 0.00 | 6.35 | 3.33 | 0.00 |

[$] The sum over all knowledge areas is higher than 100% because of the classification of the same publication in more than one area. [$$] The sum over all continents is higher than 100% because of the presence of authors from different countries in the same publication.

**Table 2.** Number of publications retrieved from the Scopus platform grouped for decades, according to the search arguments.

| Search Argument (*Cannabis + . . .* ) | 1961–1970 | 1971–1980 | 1981–1990 | 1991–2000 | 2001–2010 | 2011–2017 | Total |
|---|---|---|---|---|---|---|---|
| Biochemical | 2 | 34 | 12 | 26 | 97 | 126 | 297 |
| Biology | 2 | 7 | 5 | 16 | 82 | 98 | 210 |
| Forensic genetics | 0 | 0 | 0 | 0 | 11 | 18 | 29 |
| Genetics | 3 | 21 | 13 | 36 | 224 | 389 | 686 |
| Molecular markers | 0 | 1 | 0 | 2 | 24 | 33 | 60 |
| Traceability | 0 | 0 | 0 | 0 | 0 | 2 | 2 |
| Total | 7 | 63 | 30 | 80 | 438 | 666 | 1284 |

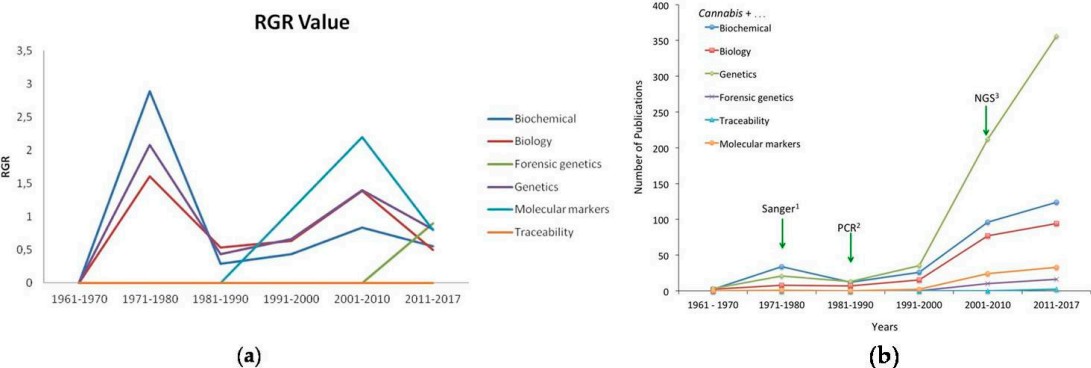

**Figure 1.** Patterns of publication on *Cannabis* recovered from the Scopus platform using six different search terms. (**a**) Relative growth rate (RGR) of world's scientific publications for each search argument. (**b**) Absolute number of publications for each search argument. Three important events related to the development of biotechnological methods are highlighted. [1] In 1977, Fred Sanger [19] described an enzyme-based methodology for DNA sequencing. [2] In 1989, Mullis and Faloona [20] proposed the Polymerase Chain Reaction (PCR) methodology for in vitro DNA amplification. [3] In early 2005, the next-generation sequencing (NGS) was introduced by Life Science through the 454-pyrosequencing methodology.

As a general trend, the correlation of the number of publications with the main advances of biotechnological tools suggests the remarkable influence of next-generation sequencing (NGS) starting in the 2000s (Figure 1b). Even though "Genetics," "Forensic genetics," and "Molecular markers" are search terms directly related to Sanger's sequencing and polymerase chain reaction (PCR) techniques, the advent of such technologies in the 1970s and 1980s, respectively, seems not to have influenced the number of publications in these periods (Figure 1b).

As expected, "Medicine" was the knowledge area with the highest number of publications for all search terms, followed by "Biochemistry, Genetics and Molecular Biology," and "Pharmacology, Toxicology and Pharmaceutics." The exception is the search term "*Cannabis* + Traceability," represented by only two publications, which were classified into "Medicine" and "Earth and Planetary Science" (Table 1).

More than 50% of the publications were authored by researchers from Europe, followed by North America for four out of the six search terms. Interestingly, Europe and Latin America presented more than 50% of publications in "Genetics" (Table 1). Countries composing the United Kingdom are responsible for the highest number of publications in Europe, while the United States is the main country in North America, Brazil in Latin America, Egypt in Africa, Australia in Oceania, and Japan in Asia.

The number of publications from Chinese researchers also needs to be highlighted, being almost the same as from Japan (Figure 2 and Figure S1). Given that the search argument "*Cannabis* + Genetics" returned the highest number of matches in the search, most of the publications were related to this search argument for all these countries, except Egypt. For Egypt, the main search argument retrieving publications was "*Cannabis* + Biochemical" (Figure 2).

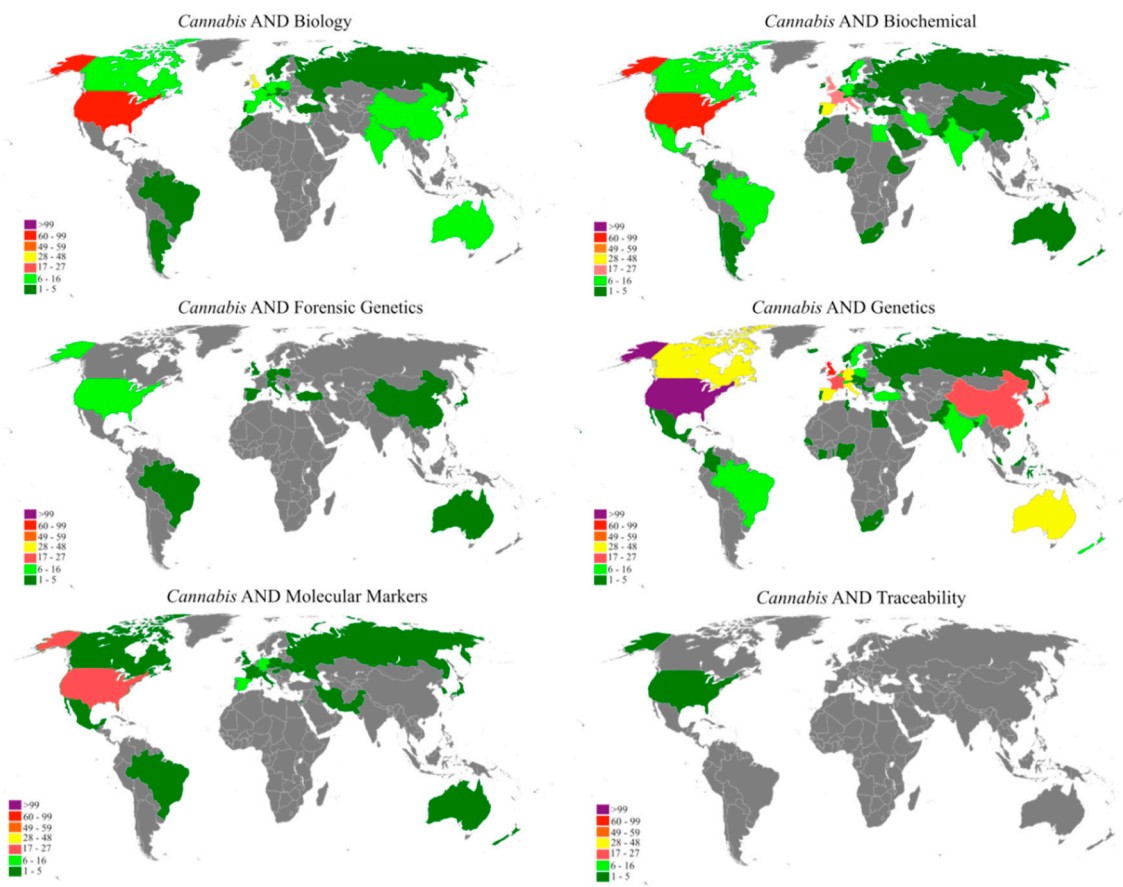

**Figure 2.** Map of the distribution of the absolute number of publications on *Cannabis* in each country as recovered from the Scopus platform using six different search arguments.

### 3.2. Variation in Results after Individual Screening

After analyzing the 1284 abstracts, or complete work when there was no abstract available on the platform, only 749 (58.3%) of the works identified by the Scopus were directly related to the *Cannabis* plant. The remaining works were mostly related to diseases, especially schizophrenia and HIV, where it was mentioned at some point in the abstract that drug use such as *Cannabis* could influence the clinical picture. Some studies reported works with illicit drugs, mainly opioids, and alcohol dependence, where the term *Cannabis* was cited as an example of a narcotic.

A general categorization of the studies involving *Cannabis* is shown in Figure 3. Using three main categories (plant, humans, and animals), the largest number of publications found was directly involving humans (56.3% of the studies), mainly studies describing the effects of the drug on the body and its chemical dependence (54.3%).

Plant studies accounted for 34.3% of the selected works, with an emphasis on studies about species and variety differentiation (13.6%) as well as for biofuel, biopolymers, and fiber or oil production (13.2%).

For the keywords "Forensic genetics," "Molecular markers," and "Traceability," the studies involving only the *Cannabis* plant were much more common than the studies involving animals and humans (92.3% for forensic genetics, 58.1% for molecular markers, and 100% for traceability). A category analysis showed that the studies returned using the keyword "Biochemical" involving plants and animals had similar results, with 58 and 59 works, respectively (Figure S2).

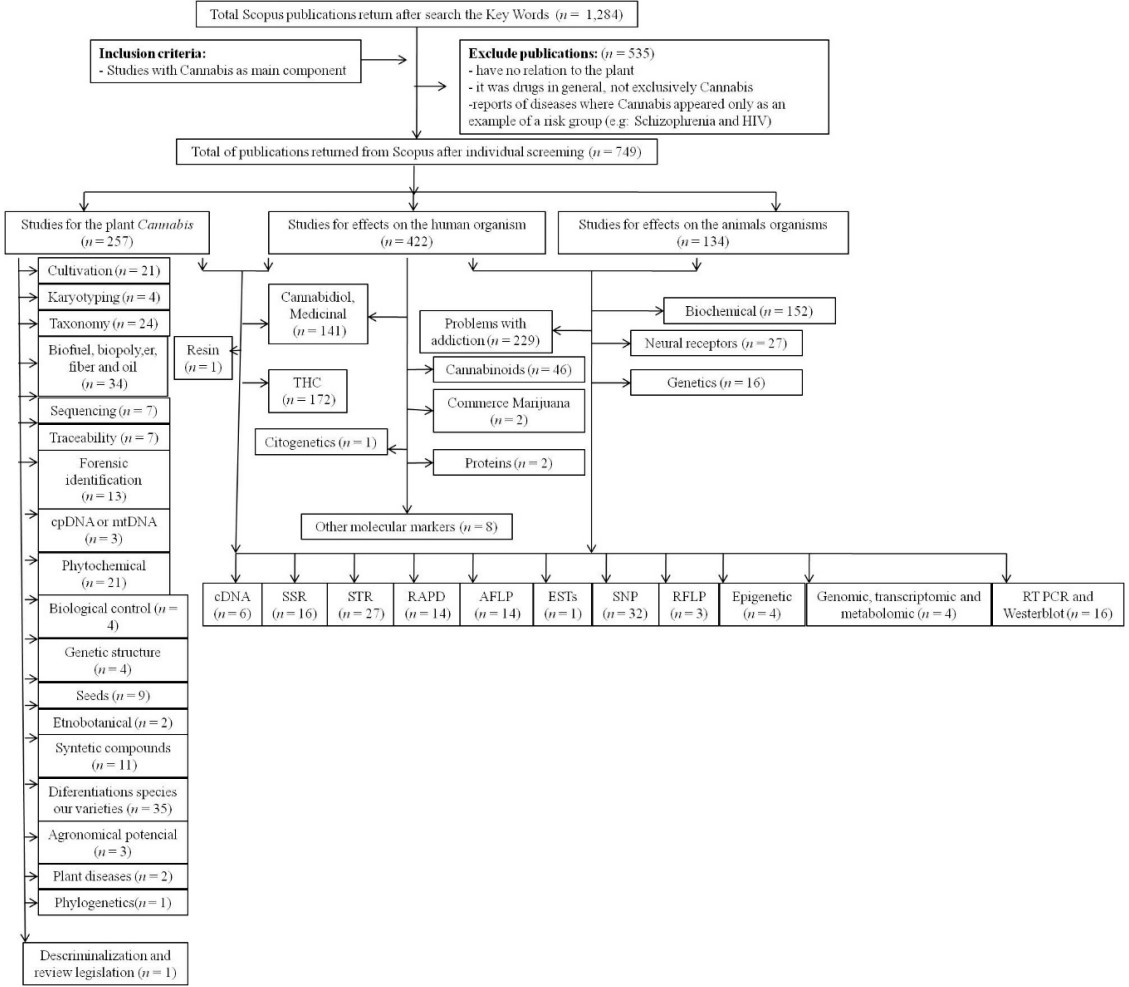

**Figure 3.** Flowchart of the scientific fields from the searched studies after the individual screening involving all search arguments employed in this study.

## 4. Discussion

The use of *Cannabis* as an illicit drug, particularly among younger people, represents potential harm to individuals and society, thus becoming a matter of public health. In addition, the occurrence of crimes related to the traffic of *Cannabis* is a problem of public security, demanding scientific strategies for forensic investigation. In particular, the use of molecular biology and genetics as tools for forensic analysis is an effective approach for the identification and traceability of *Cannabis*.

Our confidence in the conclusions of this study is limited by the assumption that the classifications related to the area of knowledge reflect the full content of the publications. In some cases, publications may present important information in secondary knowledge areas that are not reported in the online databases, since the indexing information is not recorded from the full discussion and conclusions of the paper. For example, the article published by Bataille and Bowen [21] in the journal *Chemical Geology* is indexed under "Earth and Planetary Science," but it also contains important methodological insights into the traceability of *Cannabis*.

Likewise, the geographical origin of a publication may underestimate or overestimate some countries in the case of contributions involving several institutions. In cases where students or researchers develop their studies in partner institutions with a partial PhD scholarship or as a short-term invited researcher, the authors' affiliations in a paper may not reflect the real origin of the publication.

In addition, the question of whether the authors' affiliation should be related to the institution of origin or to the institution where the study was developed remains unresolved. Nevertheless,

our results indicate the general trends of the research centers, the main interest areas, and the major gaps in studies about *Cannabis*.

The results of the present study show a significant increase in scientific research related to genetics, mainly in the last two decades. However, this increase mainly concerns aspects linked to *Cannabis* and human health, not to the genetics of *Cannabis*. After screening articles for the search term "*Cannabis* + Forensic genetics," only two studies out of the 26 retrieved by Scopus were related to humans (Figure S2).

When analyzing the results for SNP markers found with the keywords "*Cannabis* and Genetics," none of the 31 studies found was related to the genome of the plant.

Correlating this increase in scientific publications with methodological advances in DNA research [19,20,22], it seems that only the advent of next-generation sequencing (NGS) in 2005 boosted genetics studies on *Cannabis*, mainly in the last decade. Recently, NGS approaches have been employed for the complete sequencing of the *Cannabis* chloroplast [5,23] and mitochondria genomes [24], to generate whole-genome drafts [25], and for the development of SNP markers for different varieties of *Cannabis* [4].

Since the use of *Cannabis* as a psychotropic agent is an issue of public health and its trafficking is a complex problem in many countries, controlled scientific research is essential for understanding the problems and finding solutions. There are a large number of published studies reporting the medicinal and pharmacological properties of *Cannabis*, as well as research related to addiction and its psychosocial aspects. However, there are few studies related to forensic genetics and traceability (29 and two publications since 1961, respectively), research areas with crucial importance for countries where the use of *Cannabis* has restrictions and measures to prevent the traffic are needed.

Data from the Brazilian National Secretary of Drugs Policy [26] show that nine countries in South America, three in Central and North America, and 10 in Europe have legislation that discriminates against possession of all types of drugs. In the United States, the plant is considered prohibited, although medical marijuana is permitted in 20 states and in the District of Columbia, and recreational use is legalized in the District of Columbia, Washington, and California [27]. In four other U.S. states, for-profit companies have been granted licenses to produce and sell a range of products for medicinal and non-medicinal use of *Cannabis*. In Uruguay, the federal government approved legislation in the year 2013 regulating the cultivation, production, and recreational use of *Cannabis* [28].

In a survey about the trends in *Cannabis* use in 30 European and North American countries, ter Bogt and collaborators [29] reported a significant reduction in the use of *Cannabis* among 15-year-olds in the period from 2002 to 2010. However, stabilization or increase in *Cannabis* use was found, particularly in emerging countries. Although the use of *Cannabis* is higher in richer countries and in countries where it is readily available, the general trend revealed by the study was the former and heavier use of *Cannabis* in developing countries and by less affluent youths than richer and more affluent ones. Thus, the most vulnerable countries and social groups are also the most affected by the social problems related to *Cannabis* addiction, such as illegal cultivation and commercialization, violence, and psychological and cognitive health problems.

Most of the current studies on the forensic genetics of *Cannabis* are related to differentiation between psychotropic and medicinal varieties of this species. In addition to allowing varieties discrimination, molecular markers may be quite useful tools for the identification of traffic routes of illegally cultivated, distributed and commercialized *Cannabis*. Such efforts will enable more efficient prevention of the traffic in nations were *Cannabis* is prohibited.

There is enormous interest in research involving *Cannabis* and its substances, as well as in all kinds of interactions of *Cannabis* in humans. Along with the advances in the areas of genetics and molecular biology, the investigations have been growing and diversifying, but publications on forensic genetics and traceability are still limited. The problem of drugs is ongoing around the world and the discourse about *Cannabis* legalization will likely persist for many years.

Thus, a better definition of the geographic origin of plants in countries where it is prohibited is essential and, for this reason, the areas of forensic genetics and traceability emerge as primary goals. In this direction, biotechnological tools should be used for elucidation of traffic routes and for characterization of psychotropic and medicinal varieties, enabling a more efficient fight with the traffic in nations where this species has restrictions.

In general, there is a historical gap in studies on the forensic genetics and traceability of *Cannabis*. We expect that increasing interest in this issue and new biotechnological advances will lead to new studies using the genetics of *Cannabis* for forensic purposes.

**Supplementary Materials:** The following are available online at http://www.mdpi.com/2304-6775/6/4/40/s1. Figure S1: Absolute number of publications on *Cannabis* recovered from the Scopus platform using six different search arguments for each country, grouped by continents, Figure S2: Flowchart of the scientific fields from the searched studies after the individual screening involving any keywords in this work.

**Author Contributions:** Conceptualization, C.B.D.M. and V.M.S.; methodology, V.M.S.; validation, C.B.D.M., V.M.S. and R.P.M.L.; formal analysis, C.B.D.M., D.S.S. and B.J.; resources, F.A.O.C.; writing—original draft preparation, C.B.D.M.; writing—review and editing, V.M.S.; supervision, F.A.O.C. and V.M.S.; project administration, F.A.O.C. and V.M.S.; funding acquisition, F.A.O.C.

**Funding:** This research was funded by CAPES PRO-FORENSES, grant number 25/2014.

**Acknowledgments:** The authors thank CAPES by the financial support and scholarships through the project "Development of a *Cannabis sativa* L. (marijuana) traceability system in the Brazilian territory."

**Conflicts of Interest:** The authors declare no conflict of interest.

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
