# Peer review of "A Bibliometric Analysis of Cannabis Publications: Six Decades of Research and a Gap on Studies with the Plant"

_publications, doi:10.3390/publications6040040_

Round 1
Reviewer 1 Report
line 36: it is better to indicate the activity of THC as "intoxicant" and not as psycoactive because many other cannabinoids have psycoactive properties without being intoxicant
line 37: attention: THC is not the precursor of CBD! CBD and THC derive from CBGA via THCA synthase and CBDA synthase respectively.
39: not increase the activity of THC but its concentration in the plant!
Author Response
We would like to thank the referee for the comments, showing our mistakes and improving our study.
We considered all comments:
line 36: it is better to indicate the activity of THC as "intoxicant" and not as psycoactive because many other cannabinoids have psycoactive properties without being intoxicant.
We changed the expression psychoactive to “intoxicant”
line 37: attention: THC is not the precursor of CBD! CBD and THC derive from CBGA via THCA synthase and CBDA synthase respectively.
We changed the sentence, correcting this mistake
39: not increase the activity of THC but its concentration in the plant!
We changed the sentence, correcting this mistake
Reviewer 2 Report
This is an interesting contribution. Of particular interest are the correlation that has been made between the analysis of the evolution of publications and the milestones that have marked the research on the topic; the analysis of research by regions; and the categorizations made by the authors after the manual review of the content of the documents (flowcharts fig. 3 and S2: plants, humans and animals). Congratulations.
I recommend publication, but I suggest some minor changes, with the purpose of improving the final presentation of the work:
- I suggest modifying or eliminating the term "qualitative" of the title, it can produce misunderstanding, since the work is based on the analysis in numerical-statistical counts (quantitative analysis).
- Review the reference to Institute for Scientific Information, better replace it with Clarivate Analytics or in any case refer to the databases of the Web of Science.
- On page 2 it is stated that WoS and Scielo have been searched and that all the literature in these two databases is included in Scopus, has this been verified? This assertion is surprising, since although there is an important degree of overlap and Scopus is the one with the greatest coverage, there are usually specific contributions. We suggest reviewing this statement and giving the results obtained in the searches in all the databases.
- On page 3 replace "indexes" with "fields" since reference is made to search fields of the databases, not to indexes.
Author Response
Initially, we would like to thank the referee for the insightful comments that improved our manuscript.
- I suggest modifying or eliminating the term "qualitative" of the title, it can produce misunderstanding, since the work is based on the analysis in numerical-statistical counts (quantitative analysis).
We changed the term “qualitative” for “comprehensive” in the title
- Review the reference to Institute for Scientific Information, better replace it with Clarivate Analytics or in any case refer to the databases of the Web of Science.
We changed the reference “Institute for Scientific Information” for “Web of Science database” throughout the paper.
- On page 2 it is stated that WoS and Scielo have been searched and that all the literature in these two databases is included in Scopus, has this been verified? This assertion is surprising, since although there is an important degree of overlap and Scopus is the one with the greatest coverage, there are usually specific contributions. We suggest reviewing this statement and giving the results obtained in the searches in all the databases.
We not only compared the number of publications across databases, but we also verified the references. Because of these expected specific contributions, we initially performed the research in these three platforms. When we compared the references recovered during the period we searched, we found this really surprising result. After that, we decided to continue working just with the Scopus database. We highlighted in the text that specific contributions were expected, but not found.
- On page 3 replace "indexes" with "fields" since reference is made to search fields of the databases, not to indexes.
We changed the term, as suggested